# OpenOOD: Benchmarking Generalized Out-of-Distribution Detection

**Jingkang Yang**[1], **Pengyun Wang**[2,3], **Dejian Zou**[2,3], **Zitang Zhou**[2,3], **Kunyuan Ding**[2,3],
**Wenxuan Peng**[1], **Haoqi Wang**[4], **Guangyao Chen**[5], **Bo Li**[1], **Yiyou Sun**[7], **Xuefeng Du**[7],
**Kaiyang Zhou**[1], **Wayne Zhang**[4], **Dan Hendrycks**[6], **Yixuan Li**[7], **Ziwei Liu**[1] ✉

[1]S-Lab, Nanyang Technological University, Singapore
[2]Beijing University of Posts and Telecommunications, Beijing, China
[3]Queen Mary University of London, London, UK    [4]SenseTime Research, Shenzhen, China
[5]Peking University, Beijing, China    [6]University of California, Berkeley, CA, USA
[7]University of Wisconsin-Madison, Madison, WI, USA

https://github.com/Jingkang50/OpenOOD

## Abstract

Out-of-distribution (OOD) detection is vital to safety-critical machine learning applications and has thus been extensively studied, with a plethora of methods developed in the literature. However, the field currently lacks a unified, strictly formulated, and comprehensive benchmark, which often results in unfair comparisons and inconclusive results. From the problem setting perspective, OOD detection is closely related to neighboring fields including anomaly detection (AD), open set recognition (OSR), and model uncertainty, since methods developed for one domain are often applicable to each other. To help the community to improve the evaluation and advance, we build a unified, well-structured codebase called *OpenOOD*, which implements over 30 methods developed in relevant fields and provides a comprehensive benchmark under the recently proposed generalized OOD detection framework. With a comprehensive comparison of these methods, we are gratified that the field has progressed significantly over the past few years, where both preprocessing methods and the orthogonal post-hoc methods show strong potential. We invite readers to use our OpenOOD codebase to develop and contribute. The full experimental results are available in this table.

## 1 Introduction

Most existing machine learning (ML) models are trained on the closed-world assumption, where all the test data is assumed to be drawn from in-distribution (ID), *i.e.*, the same distribution as the training data [1, 2]. However, the closed-world assumption is difficult to maintain in the real world [3]. In practice, a deployed model will be inevitably exposed to unseen examples that deviated from the training distribution, which are known as out-of-distribution (OOD) samples [4, 5], which can affect ML model safety [6, 7]. While the neighborhood OOD generalization community focuses on ensuring the robustness of models to maintain high performance on OOD samples with domain shift [8], OOD detection, on the other hand, emphasizes the model reliability by requiring the identification of samples with semantic shift [5]. In other words, the goal of OOD detection is to detect samples to which the model cannot or does not want to generalize [5].

A plethora of methodologies for OOD detection has been developed in the past five years, ranging from classification-based to density-based to distance-based methods [5]. Classification-based

---

✉Corresponding author. Contact: `ziwei.liu@ntu.edu.sg`

36th Conference on Neural Information Processing Systems (NeurIPS 2022) Track on Datasets and Benchmarks.

methods take the major part of OOD detectors, which gets confidence directly from the classifier with some design that is beneficial to OOD detection [9, 10]. The design can focus on loss function [11], classifier architecture [12], and some post-hoc processing techniques [4, 13]. Specially, post-hoc methods are more suitable for real-world practice due to their plug-in simplicity without interference on the pretrained backbones that require expensive training process [14]. Density-based methods model the in-distribution with probabilistic models, which also achieve good performance and are easier for theoretical analysis [15, 16, 17]. Distance-based methods usually compute distance in the high-dimension space such as feature space and gradient space to distinguish ID and OOD [18, 19]. Some minor categories include reconstruction-based methods, which rely on the discrepancy of reconstruction-error between ID and OOD samples [20, 21].

Although more popular in the community in the past few years, there is no uniform and comprehensive benchmark to make sure the developed methods are truly effective. This brings up problems, such as some methods only reporting results on certain datasets where they are good at [22, 23]. In fact, it is normal to see that the OOD detection performance for each method varies a lot on different OOD test dataset. Also, some benchmarks are saturated with scores exceeding 99% [24], so further improvements on these benchmarks (*e.g.*, from 99.2% to 99.4%) are no longer considered valuable. Besides, even with the same benchmark, some technical details such as image preprocessing procedures [25, 26], are not unified, increasing the difficulty of a fair comparison.

Moreover, some closely related topics, such as anomaly detection (AD), open set recognition (OSR), and model uncertainty, have been developed in isolation for a long while. In fact, methods developed for OSR [27, 28], model uncertainty [29, 30], and even data augmentation methods [31] can seamlessly solve the OOD detection problem. Similarly, AD methods can apply to OOD detection task by ignoring all the labels within the in-distribution [32, 33]. Recently, a generalized OOD detection framework [5] is proposed, which unifies similar tasks such as AD, OSR, and OOD detection. Considering the inherent connections among all these neighborhood tasks, a comprehensive comparison beyond OOD detection methods is expected, so that every task can inspire each other and a joint force can be formed to promote the development of the broader model reliability community.

To address the problems, we develop a well-structured codebase called **OpenOOD**, which provides 9 benchmarks (1 from AD, 4 from OSR, 4 from OOD detection) under **the generalized OOD detection framework** [5], for comprehensive evaluation. Especially, in the OOD detection benchmarks, we systematically design different types of OOD (*i.e.*, Near-OOD & Far-OOD) for detailed analysis. All the benchmarks are carefully examined to prevent ID samples being wrongly introduced into OOD sets. Besides, we integrated 35 methods (4 from AD, 3 from OSR, 22 from OOD detection, 6 from model uncertainty plus data augmentation) using a unified, well-structured code framework in the OpenOOD, so that the majority of representative methods in all related fields can be fairly compared. In the later part, we report the comparison among all the reproduced methods across several benchmarks, followed by in-depth discussion on the results. We end up the paper with discussions on the future direction. In general, we summarize our main contributions as follows:

**Comprehensive OOD Detection Benchmarks**   We provide 9 benchmarks to comprehensively evaluate OOD detection methods. The benchmarks are systematically designed with different OOD types with careful data cleaning procedure.

**Comprehensive Comparison Across Different Tasks**   We reproduce 35 methods that are originated from OOD detection-related tasks, including AD, OSR, and model uncertainty, and compare them under the comprehensive OOD detection benchmarks.

**A Unified Codebase for OOD Detection**   We provide an open-source codebase called OpenOOD, with well-designed code structure that is able to accommodate different kinds of OOD detection methods. Codebase is available at `https://github.com/Jingkang50/OpenOOD`.

**Insights**   Through a comprehensive comparison of these methods, we share several findings: **1)** simple preprocessing methods can achieve the best score among the benchmark; **2)** Extra data seems not necessary or requires further exploration; **3)** Post-hoc methods make significant progress and are generally no worse than methods that require training.

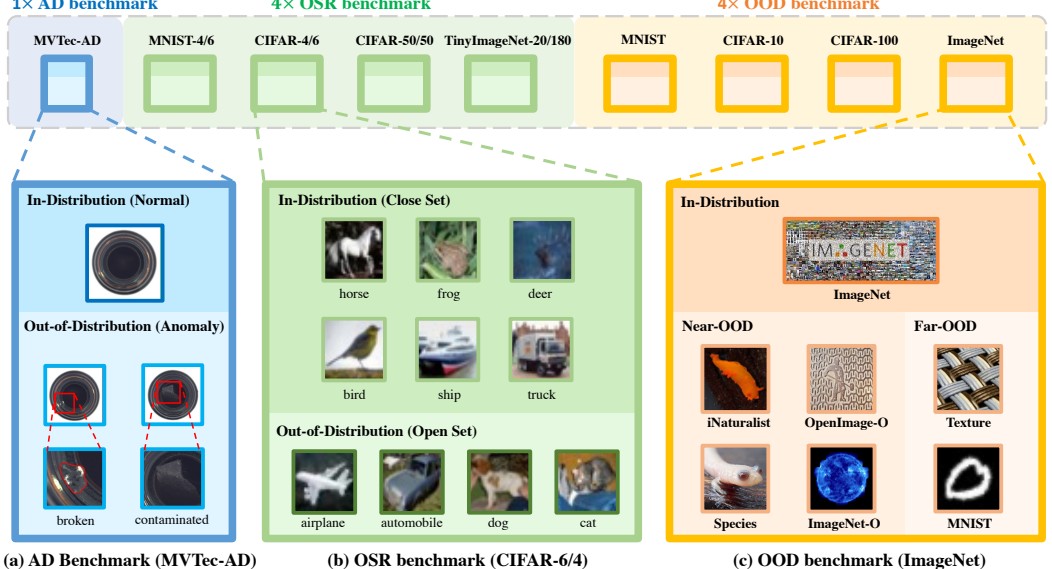

Figure 1: **Diagram of benchmarks supported by OpenOOD.** OpenOOD supports 9 benchmarks that originated from anomaly detection (AD), open set recognition (OSR), and OOD detection. Three example benchmarks are highlighted to represent AD, OSR, and OOD detection, respectively. The different benchmarking styles of AD, OSR and OOD detection clearly clearly indicate their focus. While AD requires models to be aware of the pixel-level difference like scratch, OSR and OOD detection focuses on detecting the semantic shift, where OOD (open-set) samples come from other dataset (other label-split of the dataset). All those benchmarks can be easy downloaded via this script.

## 2 Supported Tasks, Benchmarks, and Metrics

In this section, we first introduce 9 supported benchmarks in OpenOOD codebase, compassing most of the popular benchmarks across anomaly detection (AD), open set recognition (OSR), and OOD detection. Then, we introduce the metrics that are used to report the experimental results. Figure 1 shows and compares the benchmarks from different subfields.

### 2.1 Anomaly Detection

**Definition** Anomaly detection refers to the problem of finding patterns in data that do not conform to expected behavior [34]. Current anomaly detection settings often restrict the in-distribution (normality) to be with a single class [35]. According to different distribution shifts that causes the anomalies, AD tasks can be further divided into sensory AD that to detect low-level sensory anomalies, and semantic AD that to detect high-level semantic anomalies [36, 37]. However, most of the anomaly detection methods are required to address both sensory and semantic AD. The AD task can also be divided into unsupervised AD, and (semi-)supervised AD in regard to the data supervision [37].

**AD Benchmark: MVTec-AD** To evaluate methods developed for anomaly detection, we use the widely used MVTec-AD benchmark [38], which addresses the realistic industrial inspection task. MVTec-AD consists of 15 categories with 3629 images for training and validation and 1725 images for testing. While the training set only contains defect-free images, the test set contains both normal images and anomalous images with various types of defects, expecting anomaly detectors to distinguish abnormal samples from normal ones. Notice that although MVTec-AD contains 15 categories, anomaly detectors only focus on one category at one time and are trained in an unsupervised manner. Therefore, although AD methods can address OOD detection tasks by ignoring ID classes, OSR and OOD detection methods are not applicable to MVTec-AD. OpenOOD supports MVTec-AD mainly to guarantee the correctness of AD methods, and encourage more methods under the generalized OOD detection framework to be applied in all of AD, OSR, and OOD settings.

## 2.2 Open Set Recognition

**Definition**    Machine learning models trained in the closed-world setting can incorrectly classify test samples from unknown classes as one of the known categories with high confidence. Open set recognition (OSR) task is proposed to address this problem, with their own terminology of "known known classes" to represent the categories that exist at training, and "unknown unknown classes" for test categories that do not fall into any training category [39]. Formally, OSR requires a multi-class classifier to simultaneously: 1) accurately classify test samples from "known known classes", and 2) detect test samples from "unknown unknown classes".

**OSR Benchmarks**    We include 4 common-used OSR benchmarks. The common practice for building OSR benchmarks is to divide the categories of existing datasets into two parts, called closed and open set [40, 41]. Machine learning models are trained only on the closed set. When testing, the models are evaluated on the entire test set and need to separate open set samples from closed set samples. **MNIST-6/4** is based on MNIST [42] and splits the dataset into 6 classes for training and 4 classes for testing. The experiments need to run and average on 5 different splits. Similarly, **CIFAR-6/4** and **CIFAR-50/50** are constructed on CIFAR-10 [43] and CIFAR-100 [44], respectively. **TinyImageNet-20/180** splits close and open set from TinyImageNet [23].

## 2.3 Out-of-Distribution Detection

**Definition**    Out-of-distribution detection, or OOD detection, aims to detect test samples drawn from a distribution different from the training distribution, with the definition of the distribution to be well-defined according to the application in the target. For most machine learning tasks, especially image classification tasks, the distribution should refer to "label distribution", meaning that OOD samples should not have overlapping labels w.r.t. training data. Note that the training set usually contains multiple classes, and OOD detection should NOT harm the ID classification capability.

**OOD Benchmarks**    The common practice for building OOD detection benchmarks is to consider an entire dataset as in-distribution (ID), and then collect several datasets that are disconnected from any ID categories as OOD datasets. OpenOOD supports 4 OOD benchmarks, which are named after ID datasets, including MNIST [45], CIFAR-10 [43], CIFAR-100 [44], and ImageNet [2]. We further design near-OOD and far-OOD datasets to facilitate detailed analysis of the OOD detectors. Near-OOD datasets only have semantic shift compared with ID datasets, while far-OOD further contains obvious covariate (domain) shift.

**MNIST**    MNIST [42] is a 10-class handwriting digit dataset that contains 60k images for training and 10k for test. For OOD dataset, near-OOD includes NOTMNIST [46] and FashionMNIST [47], which share a similar background style with MNIST. Far-OOD consists of textural dataset Texture [48], two object datasets including CIFAR-10 [43] and TinyImageNet [2], and a scene dataset Places-365 [49]. All the far-OOD datasets have obviously different styles compared to MNIST. If applicable, we only utilize test set from OOD datasets. CIFAR-10 and Tiny-ImageNet test sets have 10k images each. Places365 contains 36.5k test images. We use the entire 5,640 Texture images.

**CIFAR-10**    CIFAR-10 [43] is a 10-class dataset for general object classification, which contains 50k training images and 10k test images. As for OOD dataset, we construct near-OOD with CIFAR-100 [44] and TinyImageNet [2]. Notice that 1,207 images are removed from TinyImageNet since they actually belong to CIFAR-10 classes [50]. Far-OOD is built by MNIST [42], FashionMNIST [47], Texture [48], and Places365 [49] with 1,305 images are removed due to semantic overlaps.

**CIFAR-100**    Another OOD detection benchmark uses CIFAR-100 [44] as in-distribution, which contains 50k training images and 10k test images with 100 classes. For OOD dataset, near-OOD includes CIFAR-10 [43] and TinyImageNet [23]. Similar to CIFAR-10 benchmark, 2,502 images are removed from TinyImageNet due to the overlapping semantics with CIFAR-100 classes [50]. Far-OOD consists of MNIST [42], FashionMNIST [47], Texture [48], and Places365 [49] with 1,305 images removed.

**ImageNet-1K**    ImageNet is a large-scale image classification dataset with 1000 classes. To build OOD dataset, we use a 10k subset of Species [51] with 713k images, iNaturalist [52] with 10k images, ImageNet-O [53] with 2k images, and OpenImage-O [54] with 17k images. All of these datasets are

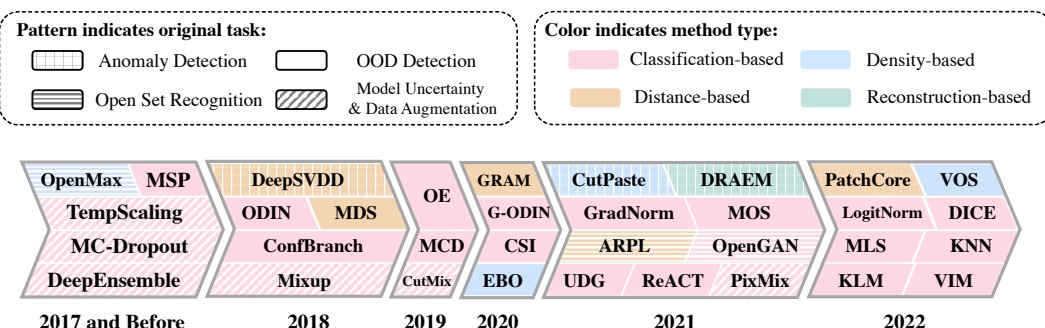

Figure 2: **Timeline and Taxonomy of Methodologies supported by OpenOOD.** OpenOOD supports 35 methods (by the time of publication) that originated from anomaly detection (AD), open set recognition (OSR), OOD detection, and model uncertainty & data augmentation. Methodologies can be categorized into classification-based, density-based, distance-based, and reconstruction-based ones.

carefully curated to get rid of samples that should belong to ID classes. Meanwhile, Texture [48], MNIST [42] and SVHN [55] are considered as far-OOD.

## 2.4 Metrics

OpenOOD mainly use the following three metrics to evaluate methods on all the supported benchmarks: **1) FPR@95** measures the false positive rate (FPR) when the true positive rate (TPR) is equal to 95%. Lower scores indicate better performance. **2) AUROC** measures the area under the Receiver Operating Characteristic (ROC) curve, which displays the relationship between TPR and FPR. The area under the ROC curve can be interpreted as the probability that a positive ID example will have a higher detection score than a negative OOD example. **3) AUPR** measures the area under the Precision-Recall (PR) curve. The PR curve is created by plotting precision versus recall. Similar to AUROC, we consider ID samples as positive, so that the score corresponds to the AUPR-In metric in some works. **In the main paper, we use AUROC as the main metric.** We provide the full results in the form of "FPR@95 / AUROC / AUPR".

## 2.5 Discussion

In this section, we discuss all 9 benchmarks that are involved in OpenOOD. The benchmark for anomaly detection can only support AD methods because their training sets do not have categorical labels. The inclusion of the AD benchmark is to guarantee the reproducibility of AD methods, and encourage more methods under the generalized OOD detection framework to be also applied to AD.

OSR and OOD detection are interchangeable in some literature due to their identical motivation to identify samples with semantic shift compared to training distribution. The only difference between them is the evaluation protocol. While OSR benchmarks are inherently difficult as datasets are split according to classes, *e.g.*, CIFAR-4/6 splits CIFAR-10 into two parts, the pretrained models turn out to be trained on the non-standard CIFAR-4 dataset. OOD detection benchmarks are designed to maintain the integrity of the training dataset, *i.e.*, using the entire CIFAR-10 for training, and introduce other datasets as OOD datasets. Some blame the early OOD benchmarks and claim that they often select easy OOD datasets and a good performance can be achieved by detecting superficial domain shift between datasets. However, recent works focus more on near-OOD, and detecting semantic shift becomes the mainstream. **The gap between OSR and OOD detection is just getting smaller.**

## 3 Supported Methodologies

In this section, we briefly introduce all 35 methods that are supported in OpenOOD. We prioritize work with open-source code for inclusion. Figure 2 lists all these methods by chronology. Different method types are marked by different colors, and different fields are marked by different patterns.

## 3.1 Methodologies for Anomaly Detection

OpenOOD supports 4 AD methods. **Deep-SVDD** [56] is a classic distance-based AD method, which enforces that the distance between a training (ID) sample and its centroid is below a certain value in the penultimate feature space. **CutPaste** [26] simply cuts out a patch from an image and pastes it back with transformation to generate anomaly samples, which further helps better density estimation for in-distribution data. **PatchCore** [57] samples ID features into a memory bank and performs the nearest neighbor searching to discover anomalies. **DRÆM** [58] is a reconstruction-based method that feeds the given image and its reconstructed version into a discriminative network to produce anomaly scores. Note that AD does not have multiple classes in the training set. To be applicable to OSR and OOD detection tasks, AD methods can simply treat all ID samples as a whole.

## 3.2 Methodologies for Open Set Recognition

OpenOOD supports 3 OSR methods. **OpenMax** [59] is the first deep learning method to address the open set problem. During inference, it replaces the classic SoftMax layer with an OpenMax layer, which fits ID samples with Weibull distribution and estimates the ID probability of test samples accordingly. **ARPL** [41] minimizes the overlap of known distributions and unknown distributions by learning discriminative reciprocal points to represent "otherness" with respect to a class. **OpenGAN** [60] uses the idea of GAN to generate negative features that are similar to external anomalous samples to enhance the open-set discriminator.

## 3.3 Methodologies for Out-of-Distribution Detection

The phenomenon of neural networks' overconfidence in out-of-distribution data attracts growing research attention over the past six years. To facilitate the comparison and reproduction, OpenOOD integrates 22 OOD detection methods in our codebase in several thriving directions:

**Post-hoc Methods**     One line of work attempts to perform post-hoc OOD detection: **MSP** [4] is the first and the most basic baseline that directly uses the maximum SoftMax score to indicate ID-ness. Later works explore other simple and more efficient indicators to distinguish ID and OOD, such as **ODIN** [13] that uses temperature scaling with gradient-based input perturbations, **MDS** [18] that measures minimum mahalanobis distance from class centroids, **EBO** [14] that uses energy-based functions, **GRAM** [61] that computes gram matrix within hidden layers, **DICE** [62] with weight sparsification in the last layer, **GradNorm** [63] that focuses on gradient statistics, **ReAct** [64] that uses rectified activation, **MLS** [51] that uses maximum logits scores rather than softmax scores, **KL-Matching** [51] that calculates the minimum KL-divergence between the softmax and the mean class-conditional distributions, **VIM** [54] that integrates both the norm of feature residual against the principal space formed by training features and the original logits to compute the OOD-ness, and **KNN** [65] that explores the efficacy of non-parametric nearest-neighbor distance for OOD detection.

All the methods above perform inference with a pretrained model in a post hoc manner, which provides several advantages including: *a) Easy-to-use*: Most OOD scoring methods are designed in a plug-and-play manner, which is simple to integrate in the existing pipeline; *b) Model-agnostic*: The testing procedure applies to a variety of model architectures, including CNNs and the recent transformer-based model ViT [66]. Moreover, the post hoc methods are agnostic to the training procedure as well, and are compatible with models trained under different losses.

**Training-time Regularization**     Another promising line of work addresses OOD detection by training-time regularization. For example, **ConfBranch** [9] builds an extra branch from the penultimate layer to estimate confidence scores. **G-ODIN** [67] decomposes the posterior to explicitly model the probability of ID-ness. **CSI** [68] explores the effectiveness of contrast learning objectives for OOD detectors (with fully unsupervised setting in this paper). **MOS** [52] uses the prior of super category to perform hierarchical OOD detection. **VOS** [69] produces better energy scores with the support of synthetic virtual outliers. **LogitNorm** [70] provides an alternative to the cross-entropy loss, which decouples the influence of logits' norm from the training procedure. Unlike post hoc methods, this line of work attempts to achieve better uncertainty estimates by training stronger models with better representations. As compensation, these methods require more computational resources.

**Training with Outlier Exposure**     Some practical works collect a bunch of external OOD samples during training to help OOD detectors to better learn ID/OOD discrepancy. Starting from **OE** [22] which encourages a flat or high-entropic prediction on given OOD samples, **MCD** [24] uses a two-branch network to enlarge branches' entropic discrepancy on OOD training data. **UDG** [50] further considers the realistic scenario where given OOD samples are not purely from OOD, so to use a clustering-based method to filter out ID samples and enhance the feature representation in an unsupervised learning manner. Although using external data is a common practice especially in industry, how to efficiently select the extra data and how to prevent the model to overfit the given OOD is still the open problem.

### 3.4   Methodologies for Model Uncertainty (including Data Augmentation Methods)

In addition to the above methods that are designed for OOD detection or OSR problems, other methods use Bayesian modeling to address model reliability problems with less-principled approximations, such as **MC-Dropout** [71] and **DeepEnsemble** [72]. Besides, **TempScaling** [73] is shown as the first and one of the simplest methods for uncertainty calibration. Some work observes that regularizing the model with data augmentation procedure will be helpful for a better estimation on uncertainty. Representative methods include **Mixup** [74], **CutMix** [75], and **PixMix** [76]. Methods that we include in this category are all require training, except temperature scaling.

## 4   Experiments

We run all the 35 methods that supported by OpenOOD, and compare them on the unified generalized OOD detection benchmarks, as shown in Table 1. This section mainly explains our systematic implementation and discussion on the results.

### 4.1   Implementation Details

To ensure the fair comparison across methods originated from different fields with different implementations, we use unified settings with common hyperparameters and architecture choices. For example, we only support LeNet [77] for benchmarks with MNIST as ID, and use ResNet-18 [78] whenever CIFAR and TinyImageNet are ID. For large-scale experiments when ImageNet is the in-distribution dataset, we use ResNet50 [78] in our benchmark comparison. If the implemented method requires training, we use the well-accepted setting with SGD optimizer, the learning rate of 0.1, the momentum of 0.9, and weight decay of 0.0005 for 100 epochs, to prevent over-tuning. If the method requires hyperparameter tuning, we only try the 5 most common values and pick the hyperparameter based on the performance of AUROC on the validation set, which is introduced in Section 2. For example, to test TempScaling [73] on ImageNet benchmark, we search the optimal temperature among 0.1, 1, 10, 100, and 1000 based on the AUROC considering ImageNet validation set (we randomly pick 10% of images from the test set) as ID, and OpenImage-O validation set as OOD. The logic behind OOD validation set selection is based on real-world practice, All these designs are for the fairness and the practicality of the comparison on the benchmark. The main benchmark development and testing are performed using 2 Nvidia RTX 3060 cards. Details of each method are listed in our GitHub wiki.

### 4.2   Main Results

**Data Augmentation is the Most Effective Method Type**     We split Table 1 vertically into several sections based on the type of method. Generally, the most effective methods lie in the category of model uncertainty works using data augmentation techniques. This group mainly contains simple and effective methods, such as the preprocessing methods such as PixMix [76] and CutMix [75]. Especially, PixMix achieves 93.1% on Near-OOD in CIFAR-10, which is the best among all the methods in this benchmark. They also ace in the most of other benchmarks. Similarly, other simple and effective methods to enhance model uncertainty estimation such as Ensemble [79] and Mixup [74] also demonstrate excellent performance.

**Extra Data Seems Not Necessary**     For the block of OOD detection, we split it into two parts based on the necessity of extra data. By comparing UDG [50] (the best from extra-data part) with KNN [65] (the best from extra data-free part), the advantage of UDG only lies in CIFAR-10 near-OOD, which

Table 1: **Main Results on Generalized OOD Detection Benchmark.** The generalized OOD detection benchmark composes 9 benchmarks from AD, OSR, and OOD detection. We denote MNIST-6/4 as M-6, CIFAR-6/4 as C-6, CIFAR-50/50 as C-50, TinyImageNet-20/180 as TIN-20 to save space. We only report the metric of AUROC. ☼ denotes methods that require training. ※ denotes post-hoc methods. + denotes methods with extra data. ⊙ means running time beyond 48 hours. "Avg." averages all the provided AUROCs within the block. "Acc." and "Time" reports the ID classification performance and the running time on the CIFAR-100 benchmark for universal comparability. Notice that this table only reports average AUROCs results for each benchmark. We also provide an Excel table to show the full experiment results, where "FPR@95 / AUROC / AUPR" is reported for each dataset in each benchmark. The "Avg." value of the OOD detection benchmark can be compared with each other only if the background color is the same.

| | AD | OSR | | | | | OOD Detection (Near-OOD / Far-OOD) | | | | | Acc. | Time |
| | MVTec | M-6 | C-6 | C-50 | T-20 | Avg. | MNIST | CIFAR-10 | CIFAR-100 | ImageNet | Avg. | | (sec.) |
|---|---|---|---|---|---|---|---|---|---|---|---|---|---|
| **- Anomaly Detection** | | | | | | | | | | | | | |
| ※ DeepSVDD [56] (ICML'18) | 90.8 | 55.8 | 48.4 | 46.4 | 52.7 | 50.9 | 54.8 / 55.0 | 56.4 / 58.9 | 53.5 / 49.1 | ⊙ | 54.9 / 54.3 | - | 1,254 |
| ☼ CutPaste [26] (CVPR'21) | 91.2 | 46.5 | **84.0** | 66.4 | 56.2 | 63.3 | **85.1 / 92.4** | **80.3 / 83.2** | 71.7 / 83.3 | ⊙ | **79.0** / 86.3 | - | 4,712 |
| ☼ DRÆM [58] (ICCV'21) | 97.0 | **57.7** | 63.4 | **72.3** | **75.1** | **67.2** | 79.3 / 99.1 | 77.3 / 83.3 | **72.7 / 85.4** | ⊙ | 76.5 / **89.3** | - | 4,452 |
| ☼ PatchCore [57] (CVPR'22) | **98.0** | ⊙ | ⊙ | ⊙ | ⊙ | ⊙ | ⊙ | ⊙ | ⊙ | ⊙ | ⊙ | ⊙ | ⊙ |
| **- OSR & OOD Detection (w/o Extra Data, w/o Training)** | | | | | | | | | | | | | |
| ※ OpenMax [59] (CVPR'16) | N/A | 97.3 | 84.2 | 70.9 | 70.1 | 80.6 | 94.2 / 98.9 | 85.8 / 86.2 | 73.1 / 63.1 | 66.0 / 84.9 | 79.8 / 83.3 | 56.1 | 473 |
| ※ MSP [4] (ICLR'17) | N/A | 96.2 | 85.3 | 81.0 | 73.0 | 83.9 | 91.5 / 98.5 | 86.9 / 89.6 | 80.1 / 77.6 | 69.3 / 86.2 | 81.9 / 87.9 | 77.1 | 77 |
| ※ ODIN [13] (ICLR'18) | N/A | 98.0 | 72.1 | 80.3 | **75.7** | 81.8 | 92.4 / 99.0 | 77.5 / 81.9 | 79.8 / 78.5 | 73.2 / 94.4 | 80.7 / 88.4 | 77.1 | 267 |
| ※ MDS [18] (NeurIPS'18) | N/A | 89.8 | 42.9 | 55.1 | 57.6 | 62.6 | **98.0** / 98.1 | 66.5 / 88.8 | 51.4 / 70.1 | 68.3 / 94.0 | 71.0 / 87.7 | 75.8 | 1,113 |
| ※ Gram [61] (ICML'20) | N/A | 82.3 | 61.0 | 57.5 | 63.7 | 66.1 | 73.9 / **99.8** | 58.6 / 67.5 | 55.4 / 72.7 | 68.3 / 89.2 | 64.1 / 82.3 | 77.1 | 234 |
| ※ EBO [14] (NeurIPS'20) | N/A | **98.1** | 84.9 | 82.7 | 75.6 | 85.3 | 90.8 / 98.8 | 87.4 / 88.9 | 71.3 / 68.0 | 73.5 / 92.8 | 80.7 / 87.1 | 77.1 | 92 |
| ※ GradNorm [63] (NeurIPS'21) | N/A | 94.5 | 64.8 | 68.3 | 71.7 | 74.8 | 76.6 / 96.4 | 54.8 / 53.4 | 70.4 / 67.2 | 75.7 / 95.8 | 69.4 / 78.2 | 77.1 | 996 |
| ※ ReAct [64] (NeurIPS'21) | N/A | 82.9 | 85.9 | 80.5 | 74.6 | 81.0 | 90.3 / 97.4 | 87.6 / 89.0 | 79.5 / 80.5 | 79.3 / 95.2 | 84.2 / 90.5 | 75.8 | 82 |
| ※ MLS [51] (ICML'22) | N/A | 98.0 | 84.8 | 82.7 | 75.5 | 85.3 | 92.5 / 99.1 | 86.1 / 88.8 | **81.0** / 78.6 | 73.6 / 92.3 | 83.3 / 89.7 | 77.1 | 57 |
| ※ KLM [51] (ICML'22) | N/A | 85.4 | 73.7 | 77.4 | 69.4 | 76.5 | 80.3 / 96.1 | 78.9 / 82.7 | 75.5 / 74.7 | 74.2 / 93.1 | 77.2 / 86.7 | 77.1 | 118 |
| ※ VIM [54] (CVPR'22) | N/A | 88.8 | 83.5 | 78.2 | 73.9 | 81.1 | 94.6 / 99.0 | 88.0 / 92.7 | 74.9 / **82.4** | 79.9 / **98.4** | 84.4 / **93.1** | 77.1 | 56 |
| ※ KNN [65] (ICML'22) | N/A | 97.5 | **86.9** | **83.4** | 74.1 | **85.5** | 96.5 / 96.7 | **90.5 / 92.8** | 79.9 / 82.2 | **80.8** / 98.0 | **86.9** / 92.4 | 77.1 | 85 |
| ※ DICE [62] (ECCV'22) | N/A | 66.3 | 79.3 | 82.0 | 74.3 | 75.5 | 78.2 / 93.9 | 81.1 / 85.2 | 79.6 / 79.0 | 73.8 / 95.7 | 78.2 / 88.3 | 76.6 | 82 |
| **- OSR & OOD Detection (w/o Extra Data, w/ Training)** | | | | | | | | | | | | | |
| ☼ ConfBranch [9] (arXiv'18) | N/A | N/A | N/A | N/A | N/A | N/A | 59.8 / 60.8 | 88.8 / 90.8 | 68.9 / 70.7 | ⊙ | 72.5 / 74.1 | 76.2 | 2,711 |
| ☼ G-ODIN [67] (CVPR'20) | N/A | N/A | N/A | N/A | N/A | N/A | 81.0 / 79.2 | 89.0 / 95.8 | 76.4 / **86.0** | ⊙ | 82.1 / 87.0 | 74.5 | 2,780 |
| ☼ CSI [68] (NeurIPS'20) | N/A | N/A | N/A | N/A | N/A | N/A | 75.8 / 91.6 | 89.1 / 92.5 | 70.8 / 66.3 | ⊙ | 78.6 / 83.5 | 61.2 | 19,716 |
| ☼ ARPL [41] (TPAMI'21) | N/A | N/A | N/A | N/A | N/A | N/A | **93.9** / 99.0 | 87.2 / 88.0 | 74.9 / 74.0 | ⊙ | 85.3 / 87.0 | 71.7 | 3,467 |
| ☼ MOS [52] (CVPR'21) | N/A | N/A | N/A | N/A | N/A | N/A | 93.2 / 94.3 | 60.8 / 61.2 | 62.8 / 55.4 | ⊙ | 72.2 / 70.3 | 63.5 | 4,026 |
| ☼ OpenGAN [60] (ICCV'21) | N/A | N/A | N/A | N/A | N/A | N/A | 42.5 / 26.7 | 36.6 / 43.2 | 69.6 / 76.0 | ⊙ | 68.8 / 73.0 | 77.1 | 2,373 |
| ☼ VOS [69] (ICLR'22) | N/A | N/A | N/A | N/A | N/A | N/A | 52.1 / 65.3 | 87.5 / 90.9 | 71.9 / 71.9 | ⊙ | 70.5 / 75.4 | 77.1 | 34,453 |
| ☼ LogitNorm [70] (ICML'22) | N/A | N/A | N/A | N/A | N/A | N/A | 91.1 / **99.4** | **92.5 / 96.7** | **78.4** / 81.3 | ⊙ | **87.3 / 92.5** | 76.5 | 8,308 |
| **- OSR & OOD Detection (w/ Extra Data, w/ Training)** | | | | | | | | | | | | | |
| + OE [22] (ICLR'19) | N/A | N/A | N/A | N/A | N/A | N/A | N/A | 76.4 / 75.2 | 63.7 / **71.0** | N/A | 70.0 / 73.1 | 51.4 | 8,122 |
| + MCD [24] (ICCV'19) | N/A | N/A | N/A | N/A | N/A | N/A | N/A | 25.7 / 25.4 | 49.7 / 33.8 | N/A | 37.7 / 29.6 | **77.8** | 6,167 |
| + UDG [50] (ICCV'21) | N/A | N/A | N/A | N/A | N/A | N/A | N/A | **91.9 / 93.4** | **75.8** / 67.7 | N/A | **83.8 / 80.5** | 77.2 | 10,110 |
| **- Model Uncertainty** | | | | | | | | | | | | | |
| ☼ MCDropout [71] (ICML'16) | N/A | 96.2 | 84.5 | 81.1 | 73.6 | 83.9 | 91.5 / 97.1 | 87.3 / 90.4 | 80.1 / 79.4 | N/A | 86.6 / 88.9 | 77.1 | 9,856 |
| ☼ DeepEnsemble [72] (NeurIPS'17) | N/A | **97.2** | **87.8** | **83.1** | **76.0** | **86.1** | **96.1 / 99.4** | **90.6 / 93.2** | **82.7** / 80.7 | N/A | **89.8 / 91.1** | **80.5** | 15,380 |
| ※ TempScale [73] (ICML'17) | N/A | 96.5 | 85.6 | 82.0 | 73.9 | 84.5 | 91.7 / 98.7 | 87.9 / 91.0 | 80.5 / **81.4** | N/A | 86.7 / 90.3 | 76.8 | 33 |
| **- Data Augmentation** | | | | | | | | | | | | | |
| ☼ Mixup [74] (ICLR'18) | N/A | 95.7 | 80.9 | **81.9** | **76.2** | 83.7 | 86.1 / 94.2 | 85.3 / 86.4 | 80.5 / 78.6 | N/A | 83.9 / 86.4 | 78.7 | 3,086 |
| ☼ CutMix [75] (ICCV'19) | N/A | **96.3** | 81.4 | 79.9 | 71.9 | 82.4 | **94.0** / 91.4 | 87.8 / 90.2 | **80.7** / 79.2 | N/A | 87.5 / 86.9 | **79.2** | 3,967 |
| ☼ PixMix [76] (CVPR'21) | N/A | 93.9 | **90.9** | 78.0 | 73.5 | **84.1** | 93.7 / **99.5** | **93.1 / 95.7** | 79.6 / **85.5** | N/A | **88.8 / 93.5** | 77.1 | 4,537 |

does not meet the expectation as a large quantity of real outlier data is required. The extra data we use in this benchmark is the entire TinyImageNet training set, which is not purely OOD. In this case, among outlier-based methods, only UDG has a reasonable performance considering other methods are not tuned to accept this kind of extra data we provide. However, the choice of training outliers can greatly affect the performance of OOD detectors, so further discussion on this topic is worth exploring.

**Post-Hoc Methods Outperform Training in General**     For OOD and OSR methods without extra data, we further split them into two parts, one that needs a training process and the other does not. Surprisingly, those methods that require training do not necessarily obtain higher performance. Generally, methods that require training do not outperform inference-only methods. Nevertheless, the trained models can be generally used in a combined way with the post-hoc methods, which could potentially further increase their performance.

**CIFAR Benchmark is NOT Easier than ImageNet Benchmark**     We find that the OOD detection performance score for the ImageNet dataset is generally higher than that of CIFAR-10 and CIFAR-100, which is another surprising discovery considering ImageNet is composed of more complex data than others and seems difficult. Admittedly, from the perspective of real-world application, solutions that perform well on higher-resolution datasets like ImageNet is more practical.

**Post-Hoc Methods are Making Progress**     In general, recent post-hoc methods have better performance compared to previous methods, suggesting the direction of inference-only methods is promising and enjoy progress. It could be found that while previous methods focus on toy datasets, recent methods improve performance on more realistic datasets. For example, the classic MDS performs well on MNIST but poorly on CIFAR-10 and CIFAR-100, and recent KNN maintains good performance on MNIST, CIFAR-10, CIFAR-100, and also shows outstanding performance on ImageNet. Notably, methods that are compatible between toy datasets (MNIST) and real-world datasets (ImageNet) have received increasing attention.

**Some AD Methods are Good at Far-OOD**     Although AD methods were originally designed to detect pixel-level appearance differences on MVTec-AD dataset, it proves to be potent when it comes to far-OOD detection such as DRAEM and CutPaste. Both methods achieved high performance on far-OOD detection, especially when using CIFAR-100 as the in-distribution dataset.

**OSR Benchmarks Aligns with OOD Detection Benchmarks**     At the beginning of the development of the OOD detection field, OOD was defined as those data that differs significantly from the in-distribution data. However, as the topic develops, the expectation of OOD detection today is to be able to discriminate class out-of-distribution samples, which is a more practical and challenging task. OSR benchmarks manually divide the categories into closed set and open set, so that the ID and OOD are differ in label distribution but with the same domain. The setting actually aligns with the near-OOD detection task, with negligible domain difference but explicit semantic shift. As the result, the experiment shows that better OSR methods usually have a better near-OOD result (*e.g.*, KNN). In sum, OSR benchmarks align with OOD detection benchmarks to a great extent.

**Comparison on ID Accuracy**     As both OSR and OOD detection methods should not downgrade the classification capability, we also report the ID classification results on the CIFAR-100 benchmark for easy comparison. The result indicates that most of the OSR/OOD methods do not affect ID classification performance.

**Model Efficiency**     We also report the entire training plus the testing time of the OOD detectors. For those methods that only require pretrained models, the training time for pretrained models is dismissed. Apart from the post-hoc methods that have minimal computational cost as expected, training with modified loss function (ConfBranch, G-ODIN, ARPL) takes a short training time but with decent performance. Data augmentation methods can help achieve great results but the computational cost is a bit higher than the aforementioned ones.

# 5  Outlook and Conclusion

In this paper, we compare over 30 methods across the fields of AD, OSR, OOD detection, and model uncertainty, under a unified generalized OOD detection benchmark. Several insights are highlighted: **1)** simple preprocessing methods can achieve the best score among the benchmark; **2)** Extra data seems not necessary or requires further exploration; **3)** Post-hoc methods make significant progress and generally outperform methods that require training. We also provide good protocols for our developers to easily integrate their methods into OpenOOD for fair and comprehensive comparisons to make substantial progress.

**Weakness**    The weakness of the benchmark results is that every method only runs one time, without multiple runs with random seeds, due to the limited computational resource. For a similar reason, we do not include training methods on large-scale ImageNet.

**ML Safety**    Out-of-distribution detection can be used to detect unexpected anomalies [7], emergent phenomena [80], unknown unknowns [6], and Black Swans [81]. Moreover, OOD detection can be used to detect malicious use or network intruders, and OOD detectors could detect when an adversary is trying to cause a system to fail. By reducing exposure to hazards, OOD detection can reduce risks and improve safety.

**Future Work**    In the future, apart from maintaining the codebase, it is promising to extend our benchmark towards more robust OOD detectors [82] and object-level OOD detection and generalization [69, 83, 84, 85, 86, 87], which provides finer-grained visual guidance in safety-critical applications, such as autonomous driving and medical image analysis, etc.

**Social Impact**    We provide a unified and comprehensive evaluation of generalized OOD detection benchmark, which guides the community to conveniently pick out the most suitable methods. Also, the release of the open-source codebase OpenOOD greatly reduces the potential redundant work, which has a favorable societal impact.

## Acknowledgments and Disclosure of Funding

This work is supported by NTU NAP, MOE AcRF Tier 2 (T2EP20221-0033), and under the RIE2020 Industry Alignment Fund – Industry Collaboration Projects (IAF-ICP) Funding Initiative, as well as cash and in-kind contribution from the industry partner(s). BL is supported by the National Research Foundation, Singapore under its AI Singapore Programme (AISG Award No: AISG2-PhD-2022-01-029). Yiyou Sun, Xuefeng Du and Yixuan Li are generously supported by a Meta Research Award.

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
