# OpenReview forum: "OpenOOD: Benchmarking Generalized Out-of-Distribution Detection"
_NeurIPS.cc/2022/Track/Datasets_and_Benchmarks — NeurIPS 2022 Datasets and Benchmarks _

### Official Review · Reviewer_8Agt · 2022-07-11
**Review for OpenOOD**

**Rating:** 8
**Confidence:** 5
**Clarity:** The paper is thoughtfully written and…

**Strengths:**

* This codebase contains 30+ OOD detection-related methods and divides these methods into categories on several dimensions. This codebase helps the researchers in this community, especially newcomers, fully get in touch with OOD detection.
* This codebase release an OpenOOD pipeline that contains many OOD detection-related benchmarks. This helps the following researchers fairly compare different methods on different tasks.
* With the help of this unified pipeline, the authors give insightful comparison results between tasks inside one single task and tasks among different tasks.

**Weaknesses:**

* This codebase lacks documents right now (at least there is no intuitive link for documentation). Nevertheless, the given code can still provide enough reproducibility.
* This codebase mainly focuses on the performance of the model and ignores the training costs, which are also crucial metrics of different methods.

**Additional Feedback:**

N/A

**Correctness:**

This submission is a benchmark, and the experiment design and evaluation are appropriate.

**Documentation:**

As referred above, this codebase lacks documentation now. Nevertheless, the given can give enough reproducibility, in my opinion. I strongly encourage authors to update the whole usage of this codebase.

**Relation To Prior Work:**

This work relates to the prior work "Generalized Out-of-Distribution Detection: A Survey," which was also developed by this team of authors. This work focuses on the codebase and experiments, yet the prior work focuses on the comprehensive survey of this field.

**Summary And Contributions:**

This paper organizes several OOD detection-related tasks, including OOD detection, AD, OSR, etc. By releasing a codebase containing methods, evaluation, and benchmark tasks, the authors undoubtedly contribute significantly to the machine learning community in an open environment. Finally, the authors comprehensively compare methods and summarize this field's progress over the past few years.

---

> ### Author Response · Authors · 2022-08-29
> **Response to Reviewer 8Agt**
>
> We thank the reviewer for the constructive suggestions and feedback. We provide discussions and explanations around the reviewer's concerns as follows.
>
> **Q1: Lack of Documentation**
> > This codebase lacks documents right now (at least there is no intuitive link for documentation). Nevertheless, the given code can still provide enough reproducibility.
>
> Thank you for pointing it out.
> We are building the tutorial of the OpenOOD benchmark (https://github.com/Jingkang50/OpenOOD/issues/80) and the detailed explanation of methods we implemented (https://github.com/Jingkang50/OpenOOD/issues/62) in the issue. While finish them all, these contents are to be moved to wiki page https://github.com/Jingkang50/OpenOOD/wiki/2.-Implemented-Methods for users' reference.
>
> **Q2: Lack of Evaluation on Training Costs**
> > This codebase mainly focuses on the performance of the model and ignores the training costs, which are also crucial metrics of different methods.
>
> Thank you for pointing it out. We have added the required time of each methods on the CIFAR-100 benchmark, which aims to address the training costs.

---

### Official Review · Reviewer_h82X · 2022-07-20
**A poor presentation of a huge research work.**

**Rating:** 6
**Confidence:** 4

**Strengths:**

Paper strengths:

 - the main strength of the paper, which corresponds in my opinion with the most important
   contribution is the open source codebase that the authors have built and released as part of this
   work. According to the authors, the code should not only allow replicating all the paper's
   results, but it should also be easily extensible in order to represent an extremely suitable
   solution to start developing new algorithms to study Generalized OOD detection tasks;
 - the presentation of a benchmark tackling OOD detection (and related fields) is certainly timely.
   Existing literature in these research fields has progressed in parallel streams without many points of
   contact;
 - the paper's motivations are well presented;
 - it is clear from the entire paper that the research work has been huge, considering various
   fields, a very large number of benchmark tracks and methodologies;
 - the proposed insights, inferred from experimental results, are quite interesting, showing that
   often more complex or more computationally expensive approaches do not provide better results in
   practice when evaluating on a large number of settings.


**Weaknesses:**

Main high level weaknesses:

 - the most important problem of this work is that the authors' target was too high, considering too
   many methods and settings. As a result, the time and space to effectively present everything were
   probably not enough. The main consequence is that the paper has obvious writing problems, and in
   many places the choices of what to describe in detail and what not to describe were wrong. Just
   to make an example many of the considered methodologies are not described well enough so that the
   reader cannot understand how they are executed (e.g. is the considered ARPL [5, 6] version the
   ECCV20 one or the TPAMI 21 one? Is the CSI [56] the completely self-supervised version or the
   supervised one? These pieces of information are quite important, given that results of these methods in various versions
   may certainly change a lot). Authors could have probably saved some space discarding some less
   useful discussions/descriptions like the detailed code presentation of Sec 4;
 - the experimental evaluation could benefit from a better and clearer organization. All the paper's
   experimental results are provided in a single very large table whose purpose is to allow easy
   comparison among all methods, but that at the same time pushes the reader to compare numbers
   which are not really comparable. For example some methods use additional auxiliary information
   and therefore should not be included among the others. This is the case for example of MOS which
   uses hierarchical categorical information. Also methods based on OE, even if they are grouped in
   a separated part of the table should not be compared with the others as the choice of the
   training time OOD data may highly influence their performance and thus the numbers reported here
   may not be really relevant. Moreover the "Avg" column is certainly misleading. Indeed even if
   the authors do not refer to it in the text the fact that it appears in the table pushes
   the reader to compare numbers for papers of different groups, which cannot be done considering
   that not all methods are tested on all benchmarks.

Other weaknesses:

 - there are some inconsistencies between the figures/tables and text. For example, section 3
   describes two methods (KL mathing, AugMix) which did not appear in the schematic organization of
   Figure 2, which on the other hand considered an unknown "KDAD" method. Moreover, while KL matching
   appears in Table 1, this is not the case for AugMix, which is substituted by CutMix;
 - the citations in Table 1 are not consistent in format with the others, it is therefore not easy
   to link a method name to the corresponding paper in the references;
 - Table 1 does not contain results for many combinations of methods and benchmarks. While there are
   undoubtedly some reasons behind these choices some of them may be not obvious to the reader, the authors
   should justify them. Why, for example, have the AD methods been run for all benchmarks, including the OSR
   and OOD detection ones, but not for ImageNet? Why are PatchCore results limited to MVTec?
 - there are some questionable choices. For example, the inclusion of the MVTec benchmark is not
   useful considering that only AD methods have been tested on it. Moreover, the methods using OOD
   data at training time have been provided with a strange choice for them: by the authors' admission
   TinyImageNet is not purely OOD and only UDG has been designed to deal with this situation, so why
   did they choose this dataset?
 - in general, the choices behind some benchmarks are not completely sound. In particular, I do not
   find it useful to consider far OOD cases in which there is a covariate shift between ID and OOD data
   in the test set, it makes the problem trivial and the results useless.
 - besides AUROC sec 2.4 describes 2 additional metrics (i.e.: FPR@95 and AUPR), where are the
   numbers for these metrics? There is no supplementary material and the github repo does not
   include them. Moreover even results for all individual experiments should be provided somewhere, for
   example with MNIST Near OOD there should be two experiments (NOTMNIST and FashionMNIST), is the
   reported number an average of them? Or have they been merged for the experiment as a single OOD
   set?

**Additional Feedback:**

I do not find the whole Sec 4 to be really useful (including figure 3). Given the limited amount of space in the paper
and even if the code is an important contribution of this research work I would have preferred a
shorter description of the implementation (maybe as part of sec 5.1) in order to save some space to
be dedicated to additional details about the tested methodologies or additional experimental
results/analyses.

Overall I'm recommending acceptance of this work, as I really think that its contribution in terms of the benchmark, research fields unification, and especially codebase are important, however I would have appreciated a better presentation.


**Clarity:**

The paper is clear, however the writing could be improved a lot. There are errors, typos and
repetitions which should be fixed.

Examples:

 - "Diagram of benchmarks that supported by OpenOOD" in the caption of Fig. 1;
 - "In the main paper, we use AUROC as the main metric in the paper." last line of sec 2.4;
 - "Methods that we include in this category are all require training, except temperature scaling"
   last line of Sec 3.4
 - "We run all the 33 methods that supported by OpenOOD" first line of sec 5
 - "Although AD methods was originally" and "OSR Benchmarks Aligns with OOD" in sec 5.2.

By the way, submissions under review should include line numbers so that the reviewers can more
easily highlight specific problems in the text.


**Correctness:**

The main correctness problem rely in the fact that, as I said above, some methods should not be
included in the same table with the others as they use additional information.

**Documentation:**

The provided code is well documented and should reproduce all experimental results.


**Ethics:**

No ethical concerns.


**Relation To Prior Work:**

The relation with prior work is clear. Previous works did not try to unify different but related research fields
under the same framework which has long be due. This paper provides this unification and effectively
allows to build a coherent picture and find out where the state of the art really is.


**Summary And Contributions:**

This paper proposes a benchmark designed to provide a quantitative analysis complement to a previous
survey paper by the same authors, which studied OOD detection and related fields to build a unified and comprehensive research framework called
Generalized Out-of-Distribution Detection.

The authors consider 9 different benchmarks from 3 related research fields (AD, OSR, OOD detection)
and evaluate the performance of over 30 methods on them, considering approaches for the
different tasks in an attempt to build a comprehensive picture of the state of the art in the
Generalized OOD detection framework.

Contributions include:

 - a large benchmark based on 9 different benchmark tracks
 - a large-scale comparison among methods designed for different but related fields. Results
   analysis allows to better understand which is the current state-of-the-art and how methods based
   on extremely different strategies compare;
 - an open source codebase which implements the tested methods and allows to easily implement
   additional ones for future research.

---

> ### Author Response · Authors · 2022-08-29
> **Response to Reviewer h28X (1/2)**
>
> We thank the reviewer for the super constructive and comprehensive suggestions and feedback. We provide discussions and explanations around the reviewer's concerns as follows.
>
> **Q1: Unclear Description of Methods**
> > The most important problem of this work is that the authors' target was too high, considering too many methods and settings. As a result, the time and space to effectively present everything were probably not enough. The main consequence is that the paper has obvious writing problems, and in many places the choices of what to describe in detail and what not to describe were wrong. Just to make an example many of the considered methodologies are not described well enough so that the reader cannot understand how they are executed (e.g. is the considered ARPL [5, 6] version the ECCV20 one or the TPAMI 21 one? Is the CSI [56] the completely self-supervised version or the supervised one? These pieces of information are quite important, given that results of these methods in various versions may certainly change a lot). Authors could have probably saved some space discarding some less useful discussions/descriptions like the detailed code presentation of Sec 4.
>
> Thank you for your detailed comments on this work! We have corrected the parts mentioned:
> - ARPL uses the method from the PAMI version, which is the extended version of RPL.
> - CSI follows the fully unsupervised manner.
>
> We totally agree that the implementation details are important for benchmarking many methods, therefore, we are building the tutorial of the OpenOOD benchmark (https://github.com/Jingkang50/OpenOOD/issues/80) and the detailed explanation of methods we implemented (https://github.com/Jingkang50/OpenOOD/issues/62) in the issue. While finish them all, these contents are to be moved to wiki page https://github.com/Jingkang50/OpenOOD/wiki/2.-Implemented-Methods for users' reference.
>
> **Q2: Better Organization of Main Results is Needed**
> > The experimental evaluation could benefit from a better and clearer organization. All the paper's experimental results are provided in a single very large table whose purpose is to allow easy comparison among all methods, but that at the same time pushes the reader to compare numbers which are not really comparable. For example some methods use additional auxiliary information and therefore should not be included among the others. This is the case for example of MOS which uses hierarchical categorical information. Also methods based on OE, even if they are grouped in a separated part of the table should not be compared with the others as the choice of the training time OOD data may highly influence their performance and thus the numbers reported here may not be really relevant. Moreover the "Avg" column is certainly misleading. Indeed even if the authors do not refer to it in the text the fact that it appears in the table pushes the reader to compare numbers for papers of different groups, which cannot be done considering that not all methods are tested on all benchmarks.
>
> Thank you for the suggestion. We provide the huge Table 1 mainly to help readers to compare all the methods across the sub-field. Although different methods might have different settings and the comparison might not be fair, we believe to compare them altogether in one table can help the community to know the influence of those "unfair effects", and some counterintuitive findings such as "training-based methods might not exceed the post-hoc ones" might intrigue the community to rethink. Admittedly, the "Avg." values are sometimes misleading. In the revised version, we reorganize Table 1, mainly to have "Avg." for OSR and OOD benchmarks, respectively. We also color-code the table, so that methods with the same color can compare their "Avg." value. We hope this revision could help the table to be easy-read.
>
>
> **Q3: Inconsistency between Figures/Tables and Text**
> > There are some inconsistencies between the figures/tables and text. For example, section 3 describes two methods (KL mathing, AugMix) which did not appear in the schematic organization of Figure 2, which on the other hand considered an unknown "KDAD" method. Moreover, while KL matching appears in Table 1, this is not the case for AugMix, which is substituted by CutMix;
>
> Thank you for pointing them out. We have fixed them accordingly.
>
> **Q4: Problem of Citation Format in Table 1**
> > The citations in Table 1 are not consistent in format with the others, it is therefore not easy to link a method name to the corresponding paper in the references;
>
> Thanks for pointing it out. We have fixed it accordingly.

---

> > ### Author Response · Authors · 2022-08-29
> > **Response to Reviewer h28X (2/2)**
> >
> > **Q5: Lack of Combination Results in Table 1**
> > > Table 1 does not contain results for many combinations of methods and benchmarks. While there are undoubtedly some reasons behind these choices some of them may be not obvious to the reader, the authors should justify them. Why, for example, have the AD methods been run for all benchmarks, including the OSR and OOD detection ones, but not for ImageNet? Why are PatchCore results limited to MVTec?
> >
> > Thank you for pointing it out, and we have added a comprehensive instruction in the Table 1 caption. Basically, many missing numbers are because the designed methods are too slow on some benchmark, especially for the AD methods. The missing value could urge the community to take computational cost into consideration. But of course, we are keeping updating and maintain the codebase, and the results will be included when we have more resources.
> >
> > **Q6: Questionable Choices in MVTec and OOD w/ Extra Data**
> > > There are some questionable choices. For example, the inclusion of the MVTec benchmark is not useful considering that only AD methods have been tested on it. Moreover, the methods using OOD data at training time have been provided with a strange choice for them: by the authors' admission TinyImageNet is not purely OOD and only UDG has been designed to deal with this situation, so why did they choose this dataset?
> >
> > We selected MVTec benchmark mainly to keep the completeness of our generalized OOD detection benchmark. As OpenOOD includes AD methods, the first reason to include MVTec is to run on the benchmark to make sure the reproductivity. Another reason is that current OSR and OOD methods are not applicable to AD benchmark, the blanks in the table is also inviting the OOD/OSR community to develop some more comprehensive and universal methods to well solve all the 9 benchmarks.
> >
> >
> > **Q7: Questionable Choices of Far-OOD**
> > > In general, the choices behind some benchmarks are not completely sound. In particular, I do not find it useful to consider far OOD cases in which there is a covariate shift between ID and OOD data in the test set, it makes the problem trivial and the results useless.
> >
> > While intuitively, OOD detection models should be much better at far-OOD, and near-OOD could be more challenging, sometimes it is not the case. By referring to the results in the OOD benchmark "Avg." in Table 1, the gap between near-OOD and far-OOD is not so large, and the far-OOD is not saturated yet, making the far-OOD problem not trivial yet. Especially, for OE-based OOD detectors, near-OOD could be easier than far-OOD, as the near-OOD-alike OE could help the detectors to better recognize near-OOD while the far-OOD detection capability could be weak. In fact, far-OOD could be challenging as the covariate shift could introduce some (magically) superious features that fool the model to consider them to be ID, and these "hard far-ood" could be another interesting thing to explore for the community.
> >
> >
> > **Q8: Full Experiment Results Should Be Provided**
> > > Besides AUROC sec 2.4 describes 2 additional metrics (i.e.: FPR@95 and AUPR), where are the numbers for these metrics? There is no supplementary material and the github repo does not include them. Moreover even results for all individual experiments should be provided somewhere, for example with MNIST Near OOD there should be two experiments (NOTMNIST and FashionMNIST), is the reported number an average of them? Or have they been merged for the experiment as a single OOD set?
> >
> > Thank you for catching it out. We provide the full experiment result using [this Google Doc](https://docs.google.com/spreadsheets/d/1gGHpdA3sSgfGpsrDUgt9lejvIbf5yOrDyJ511zsn1uY/edit?usp=sharing), where FPR@95 / AUROC / AUPR are reported for each dataset in each benchmark.
> > We also provide the link in the Table 1's caption for users' reference.

---

> > > ### Comment · Reviewer_h82X · 2022-09-02
> > > **Final Rating**
> > >
> > > The authors addressed some of my concerns, however, the paper still shows some weaknesses.
> > > In particular, I find it unacceptable that more than one and a half months after the submission of the first version of the paper the results are still incomplete. The submission of the first version with reported numbers obtained by a single run of each experiment was already quite strange. After so much time and having the possibility to revise the manuscript I would have expected the authors to perform at least two other runs in order to provide numbers you can rely on. Besides this, the main table still shows some watch placeholders for methods requiring more than 48 hours to run. It would be better to still run these experiments and highlight their time constraint separately so that the reader could still compare their performance with the competitors.
> > >
> > > As a result, I'm keeping my original rating of 6 as I think the work is valuable but it is not really complete.

---

### Official Review · Reviewer_cq3M · 2022-07-21
**Review for OpenOOD: Benchmarking Generalized Out-of-Distribution Detection**

**Rating:** 4
**Confidence:** 4
**Correctness:**

**Strengths:**

The ability of a model to recognise data samples that deviate from the training distribution is an important real-world problem. The research community is in need of a unifying benchmark dataset that can reliably assess the performance of OOD methods and that can serve as a standardised evaluation tool. The authors evaluate a large number of well-selected methods from various domains.


**Weaknesses:**

1. The majority of included datasets are of very low-resolution (i.e. MNIST, CIFAR10/100, TinyImageNet). In real-world applications, most classification problems do not operate on such low-resolution inputs. Hence, it is not clear how well the results obtained on this benchmark transfer to many other real-world problems.

2. The benchmark only covers a few very specific settings for which no motivation is given. For example, MNIST and CIFAR 10 are split into ratios of 6/4, but there is no clear motivation for doing so in the paper.

3. The description of the codebase structure is insufficiently detailed. Key components in the pipeline in Figure 3, such as the trainer component, are only roughly outlined in the paper. Some of the displayed components, e.g., the MOSTrainer, OETrainer, etc. are not mentioned in the paper. This may be due to space limitations, but then I would suggest to remove these descriptions altogether.

**Additional Feedback:**



**Clarity:**

The paper is written in an understandable way. However, there are still numerous grammar issues. When fixed, this would improve the readability of the paper.

**Documentation:**



**Ethics:**



**Relation To Prior Work:**



**Summary And Contributions:**

The paper proposes a new benchmark for out-of-distribution detection that generalizes over datasets and methods from the anomaly detection, open set recognition, and out-of-distribution detection communities. They construct a number of  datasets from existing image classification databases and evaluate around 30 methods.

---

> ### Author Response · Authors · 2022-08-29
> **Respond to Reviewer cq3M**
>
> **Q1: The Majority are Low-Resolution Datasets**
> > The majority of included datasets are of very low-resolution (i.e. MNIST, CIFAR10/100, TinyImageNet). In real-world applications, most classification problems do not operate on such low-resolution inputs. Hence, it is not clear how well the results obtained on this benchmark transfer to many other real-world problems.
>
> Thank you for pointing out this problem. Indeed, current OOD detection community is still using low-resolution images for experiments, especially before 2021. However, even after 2021, only few paper considers ImageNet-level large-scale experiments. While most of the methods did experiments on low-resolution images, the OpenOOD benchmark is supposed to track most of the existing methods on the classic benchmarks, while trying to push the community into using large-scale benchmark by including a **well-designed** ImageNet benchmark. We hope the OpenOOD followers could achieve great results regardless of the image resolution.
>
>
> **Q2: No Clear Motivation for OSR Benchmarks**
> > The benchmark only covers a few very specific settings for which no motivation is given. For example, MNIST and CIFAR-10 are split into ratios of 6/4, but there is no clear motivation for doing so in the paper.
>
> Thank you for the question. Actually, OSR benchmarks strictly follow the standard experimental settings of open set recognition works. The logic of the classic OSR benchmark is that: OSR notices that machine learning models trained in the closed-world setting can incorrectly classify test samples from unknown classes as one of the known categories with high confidence, so OSR aims to ensure the multi-class classifier to simultaneously: 1) accurately classify test samples from “known classes”, and 2) detect test samples from “unknown unknown classes”. One simple way to define “known classes” and “unknown classes” is to pick 6 classes from MNIST/CIFAR-10 as “known classes”, and to consider the rest 4 classes as “unknown classes”. The key is that the 6/4 ratios are class-wise.
>
>
> **Q3: Unclear / Unnecessary Description of Codebase Structure**
> > The description of the codebase structure is insufficiently detailed. Key components in the pipeline in Figure 3, such as the trainer component, are only roughly outlined in the paper. Some of the displayed components, e.g., the MOSTrainer, OETrainer, etc. are not mentioned in the paper. This may be due to space limitations, but then I would suggest to remove these descriptions altogether.
>
> Thank you for the suggestions, and we remove the confusing codebase structure part to highlight the experimental part.

---

### Official Review · Reviewer_cvXB · 2022-07-27
**Generally Good Work for OOD Task**

**Rating:** 6
**Confidence:** 4
**Correctness:** The claims made in this paper are cor…
**Clarity:** Yes

**Strengths:**

Algorithms
- The authors propose a comprehensive benchmark called OpenOOD, which intergrates more than 30 cutting-edge OOD algorithms. Collecting so many DL-based methods could be served as a contribution for the OOD community, where both researchers and practitioner can quickly implement algorithms on the newcomming dataset and compare their performance.

Datasets
- The authors propose 9 datasets, i.e., one anomaly detection (AD) dataset, four open set recognition  (OSR) datasets and four OOD datasets), these datasets could be regarded as a good testbed for evaluating newly proposed OOD methods.

Framework
- OpenOOD is fully integrated as a pipeline for providing more convinient task processing. The OpenOOD pipeline include Dataloader——Preprocessor——Network——Trainer——Evaluator——Postprocessor——Tools, where each steps include the currently popular OOD methods, such as the GradNorm method in the Postprocessor step.

Experimental Results
- Based on the OpenOOD, the authors draw several interesting conclusions in Section 5.2. For example, "Extra Data Seems Not Necessary" is superies but meaningful observation, since collecting additional datasets for OOD problem could be laborious.

**Weaknesses:**

The weaknesses of this paper are mainly two points:
- The computational cost makes the model runtime only once (mentioned in the Weakness subsection), this could lead to a certain extent of randomness in the current experimental results. Besides, statistical test should be considered for model comparison, not just the absolute difference of model performance.
- Currently the experimental results tell the readers: which OOD algorithms are better for different OOD tasks? The authors could provide more perspectives about: which OOD algorithms are better for a specific type of data? Besides, the authors could further provide some explainable results for understanding the inherent architecture of different OOD algorithms.

**Additional Feedback:**

No

**Documentation:**

Yes. OpenOOD provides open-source datasets and codes for the OOD task / community.

**Relation To Prior Work:**

Yes

**Summary And Contributions:**

This paper compares more than 30 algorithms on 9 datasets, including one anomaly detection (AD) dataset, four open set recognition  (OSR) datasets and four OOD datasets. Based on comprehensive experiments, the authors draw several insightful conclusions, which are regard to be helpful for the OOD community.

---

> ### Author Response · Authors · 2022-08-29
> **Response to Reviewer cvXB**
>
> **Q1: Multiple Run for Each Experiment**
> > The computational cost makes the model runtime only once (mentioned in the Weakness subsection), this could lead to a certain extent of randomness in the current experimental results. Besides, statistical test should be considered for model comparison, not just the absolute difference of model performance.
>
> We admit the limitation due to our limitation on the computational resources. However, most of the results have stable results, especially for post-hoc methods. While for methods that require training, we are keeping running and update the experiments to provide more stable results along with the codebase maintenance.
>
>
> **Q2: More Discussion on Results**
> > Currently the experimental results tell the readers: which OOD algorithms are better for different OOD tasks? The authors could provide more perspectives about: which OOD algorithms are better for a specific type of data? Besides, the authors could further provide some explainable results for understanding the inherent architecture of different OOD algorithms.
>
> Thank you for the nice suggestions. We have added the additional discussion in the Sec. 4.2 Main Results section.

---

### Official Review · Reviewer_xhpU · 2022-07-27
**Review for OpenOOD: Benchmarking Generalized Out-of-Distribution Detection**

**Rating:** 6
**Confidence:** 4
**Correctness:** The nine benchmarks are well evaluate…
**Clarity:** The paper is well-written and easy to…

**Strengths:**

- 9 OOD detection benchmarks are provided in the codebase OpenOOD.
- The proposed codebase supports 33 methods originated from AD, OSR, OOD detection and model uncertainty.
- The OpenOOD abstracts a unified pipeline to meet the requirements of different types of methods.
- Extensive experiments are conducted to obtain some interesting results about model uncertainty, extra data, Post-Hoc etc.
- The source codes and dataset have be released.


**Weaknesses:**

- The metrics of FPR@5 and AUPR are introduced in the paper, but the experimental results in terms of these metrics are not reported.  Are their results same as ones with  AUROC?
-  Although the codebase contains 33 methods, but it seems to be not enough to discuss on the characteristics of these methods according to experimental results.
- In the experiments, every method only run one time.



**Additional Feedback:**

The benchmarks include four types of methods, including classification-based, density-based, distance-based and reconstruction-based. If possible, it would be better to discuss the advantages of the four categories according to the experimental results, respectively.


**Documentation:**

The proposed codebase has been open source and includes sufficient detail for reproducibility.

**Ethics:**

There is no ethical concern in this paper.

**Relation To Prior Work:**

The related work has been systematically discussed, and the contributions of this paper are obviously different compared with existing researches.

**Summary And Contributions:**

- This paper proposes an open-source unified OOD detection codebase with 9 benchmarks.
- This paper makes comprehensive comparison across different OOD detection tasks, including anomaly detection , open set recognition and model uncertainty.
- This paper obtains some valuable findings using the proposed codebase.

---

> ### Author Response · Authors · 2022-08-29
> **Response to Reviewer xhpU**
>
> **Q1: Missing Results of FPR@95 and AUPR**
> > The metrics of FPR@95 and AUPR are introduced in the paper, but the experimental results in terms of these metrics are not reported. Are their results same as ones with AUROC?
>
> Thank you for catching it out. We provide the full experiment result using [this Google Doc](https://docs.google.com/spreadsheets/d/1gGHpdA3sSgfGpsrDUgt9lejvIbf5yOrDyJ511zsn1uY/edit?usp=sharing), where FPR@95 / AUROC / AUPR are reported for each dataset in each benchmark.
> We also provide the link in the Table 1's caption for users' reference.
>
> **Q2: More Discussion on Results**
> > Although the codebase contains 33 methods, but it seems to be not enough to discuss on the characteristics of these methods according to experimental results.
> > If possible, it would be better to discuss the advantages of the four categories according to the experimental results, respectively.
>
> Thank you for the nice suggestions. Main Results (Sec. 4.2) highlight several key findings accoring to the experimental results, which include:
> - Data Augmentation Results is the most effective method type
> - Extra data seems not necessary
> - Post-Hoc Methods outperform training-based methods in general
> - Post-Hoc Methods are making progress in the past few years
> - Some AD methods are good at Far-OOD
> - Analysis on ID accuracy and computational costs
>
> **Q3: Multiple Run for Each Experiment**
> > In the experiments, every method only run one time.
>
> We admit the limitation due to our limitation on the computational resources. However, most of the results have stable results, especially for post-hoc methods. While for methods that require training, we are keeping running and update the experiments to provide more stable results along with the codebase mantainance.

---

### Official Review · Reviewer_pgHB · 2022-07-27
**Good Benchmark, but some weakness**

**Rating:** 6
**Confidence:** 5
**Correctness:** Yes.
**Clarity:** Well-written.

**Strengths:**

(1) Overall a well-written paper and easy to follow.

(2) The paper introduces a benchmark for three related research topics, including AD, OSR, and OOD, facilitating a unified comparison of AD, OSR and OOD algorithms.

**Weaknesses:**

(1) I suggest to include a specific definition of AD, OSR and OOD and a detailed discussion on their similarities/differences before introducing the benchmark.

(2) Compared to the OOD methods, algorithms in AD and OSR are not updated. Specifically, knowledge-distillation based methods such as MKD (CVPR'21) [2]  and reverse distillation (CVPR'22) [2] have reached SOTA performance in AD. Similarly, a task-adaptive negative class envision method (CVPR'22) [3] is proposed to tackle OSR. I highly suggest to include these methods in the benchmark.

(3) I doubt that Cifar-10 and Cifar-100 could be used to build a near-OOD benchmark. Cifar-10 and Cifar-100 contain identical image content but with different label information. In my opinion, it doesn't follow the definition of OOD.

(4) It is good to have a public open-source benchmark. But it neither provides instructions for benchmark usage nor presents demos for experimental result reproduction. It is highly suggested to include specific documentation on the benchmark usage.

Ref:

[1] Mohammadreza et al., Multiresolution Knowledge Distillation for Anomaly Detection, CVPR 2021.

[2] Hanqiu et al., Anomaly Detection via Reverse Distillation from One-Class Embedding, CVPR 2022.

[3] Shiyuan er al., Task-Adaptive Negative Class Envision for Few-Shot Open-Set Recognition, CVPR 2022.


**Additional Feedback:**

Please refer to the weakness section. Thanks.

**Documentation:**

It is good to have a public open-source benchmark. But it neither provides instructions for benchmark usage nor presents demos for experimental result reproduction. It is highly suggested to include specific documentation on the benchmark usage.


**Ethics:**

It seems there is no particular ethical issue.

**Relation To Prior Work:**

Yes.

**Summary And Contributions:**

This paper introduces a benchmark for three related research topics, including AD, OSR, and OOD. Experimental settings are specified for each topic. In total, 31 methods are evaluated. This work facilitates a unified comparison of AD, OSR and OOD algorithms.

---

> ### Author Response · Authors · 2022-08-10
> **Question about CIFAR-10/100**
>
> Thank you so much for providing the detailed suggestions to help make OpenOOD better. We are currently revising the paper accordingly.
>
> However, I have a question in regard to your comments on CIFAR-10/100.
> > I doubt that Cifar-10 and Cifar-100 could be used to build a near-OOD benchmark. Cifar-10 and Cifar-100 contain identical image content but with different label information. In my opinion, it doesn't follow the definition of OOD.
>
> May I ask could you elaborate on the claim that "Cifar-10 and Cifar-100 contain identical image content but with different label information."? To make it more clear, we consider the **definition of OOD** in our paper is "samples with different semantic labels". We check the labels of CIFAR-10 and CIFAR-100 in https://www.cs.toronto.edu/~kriz/cifar.html, and we did not find the same labels.
> We find that CIFAR-100's "bus, pickup truck" and CIFAR-10's "truck" might be similar, but we did not find identical images when checking them. In fact, from Alex Krizhevsky's thesis: "The CIFAR-10 set has 6000 examples of each of 10 classes and the CIFAR-100 set has 600 examples of each of 100 **non-overlapping** classes" (https://www.cs.toronto.edu/~kriz/learning-features-2009-TR.pdf).
>
> Could you help provide more information about this statement so that we could improve our benchmark accordingly?
>
> Thank you so much!

---

> > ### Comment · Reviewer_pgHB · 2022-08-20
> > **About CIFAR10/100**
> >
> > To clarify the question, CIFAR-10 contains images from the categories of "automobile" and "truck" and CIFAR-100 subdivides the automobile into 10 classes. These classes highly overlap in terms of semantic representation. I doubt directly using the original CIFAR-10 and CIFAR-100 image sets as OOD benchmark makes sense.

---

> ### Author Response · Authors · 2022-08-29
> **Response to Reviewer pgHB**
>
> We thank the reviewer for the constructive suggestions and feedback. We provide discussions and explanations around the reviewer's concerns as follows.
>
> **Q1: Discussion on AD, OSR, OOD Definition**
> > I suggest to include a specific definition of AD, OSR and OOD and a detailed discussion on their similarities/differences before introducing the benchmark.
>
> Thank you for your suggestion, and we have added the definition of AD, OSR, and OOD with brief discussion in the beginning of Section 2: Supported Tasks, Benchmarks, and Metrics.
>
> **Q2: Algorithms in AD and OSR are not updated.**
> > Compared to the OOD methods, algorithms in AD and OSR are not updated. Specifically, knowledge-distillation based methods such as MKD (CVPR'21) [2] and reverse distillation (CVPR'22) [2] have reached SOTA performance in AD. Similarly, a task-adaptive negative class envision method (CVPR'22) [3] is proposed to tackle OSR. I highly suggest to include these methods in the benchmark.
>
> Thank you for your suggestion. We have added the implementation for reverse distillation (CVPR'22) [2] to ensure the AD methods in OpenOOD is up-to-date. Please check the RD4AD results in Table 1.
>
> **Q3: Concern about CIFAR-10 and CIFAR-100**
> > I doubt that Cifar-10 and Cifar-100 could be used to build a near-OOD benchmark. Cifar-10 and Cifar-100 contain identical image content but with different label information. In my opinion, it doesn't follow the definition of OOD. To clarify the question, CIFAR-10 contains images from the categories of "automobile" and "truck" and CIFAR-100 subdivides the automobile into 10 classes. These classes highly overlap in terms of semantic representation. I doubt directly using the original CIFAR-10 and CIFAR-100 image sets as OOD benchmark makes sense.
>
> We appreciate your question and clarification. We agree with the reviewer's definition of OOD detection that OOD samples are **semantically different** from the training (in-distribution) classes. Based on this consensus and the claim that "The CIFAR-10 set has ... 10 classes and the CIFAR-100 set has ... 100 **non-overlapping** classes" from Alex Krizhevsky's [original CIFAR paper]([https://www.cs.toronto.edu/~kriz/learning-features-2009-TR.pdf](https://www.cs.toronto.edu/~kriz/learning-features-2009-TR.pdf)), we believe CIFAR-10 and CIFAR-100 is suitable to be considered as OOD mutually.
> To be more specific, as the reviewer may concern that the "automobile" class in CIFAR-10 overlaps vehicle classes in CIFAR-100, CIFAR homepage actually claims that [*"the classes are completely mutually exclusive. There is no overlap between automobiles and trucks. "Automobile" includes sedans, SUVs, things of that sort. "Truck" includes only big trucks. Neither includes pickup trucks."*](https://www.cs.toronto.edu/~kriz/cifar.html#:~:text=The%20classes%20are%20completely%20mutually%20exclusive.%20There%20is%20no%20overlap%20between%20automobiles%20and%20trucks.%20%22Automobile%22%20includes%20sedans%2C%20SUVs%2C%20things%20of%20that%20sort.%20%22Truck%22%20includes%20only%20big%20trucks.%20Neither%20includes%20pickup%20trucks.).
> According to the classes listed below, CIFAR-10 has 4 vehicle classes and CIFAR-100 has 10 vehicle classes, but they are covering different semantics by design. We notice that the closest pair is "truck" v.s. "pickup truck", while the CIFAR homepage explicitly claims CIFAR-10's "truck" does not include CIFAR-100's "pickup truck", as we mentioned before.
> | Dataset | Vehicle Classes |
> |    --   |    --   |
> | CIFAR-10 | airplane, automobile, ship, truck |
> | CIFAR-100 | bicycle, bus, motorcycle, pickup truck, train, lawn-mower, rocket, streetcar, tank, tractor|
>
>
>
> **Q4: Benchmark Documentation**
> > It is good to have a public open-source benchmark. But it neither provides instructions for benchmark usage nor presents demos for experimental result reproduction. It is highly suggested to include specific documentation on the benchmark usage.
>
> Thank you for pointing it out.
> We are building the tutorial of the OpenOOD benchmark (https://github.com/Jingkang50/OpenOOD/issues/80) and the detailed explanation of methods we implemented (https://github.com/Jingkang50/OpenOOD/issues/62) in the issue. While finish them all, these contents are to be moved to wiki page https://github.com/Jingkang50/OpenOOD/wiki/2.-Implemented-Methods for users' reference.

---

### Meta-Review · Area_Chair_7WzS · 2022-09-11

**Recommendation:** Accept
**Confidence:** 5

**Metareview:**

All the reviews are mostly aligned, in recognition of the merits and contribution of this paper, and its potential impacts to the community.
Therefore, I would like to recommend acceptance. Even though, I still recommend authors to make further enhancements and include the response changes in this paper.

---

### Decision · Program_Chairs · 2022-09-16

Accept